# Atmospheric Boundary Layer in the Atlantic: the desert dust impact

Ioanna Tsikoudi<sup>1,2</sup>, Eleni Marinou<sup>1</sup>, Maria Tombrou<sup>2</sup>, Eleni Giannakaki<sup>2</sup>, Emmanouil Proestakis<sup>1</sup>, Konstantinos Rizos<sup>1</sup>, Ville Vakkari<sup>3,4</sup>, Holger Baars<sup>5</sup>, Annett Skupin<sup>5</sup>, Ronny Engelmann<sup>5</sup>, Zhenping Yin<sup>5</sup>, and Vassilis Amiridis<sup>1</sup>

Correspondence: Ioanna Tsikoudi (jtsik@noa.gr)

Abstract. We investigate the dynamics of the atmospheric Boundary Layer (BL) over the Atlantic Ocean, with a focus on the region surrounding Cabo Verde during the Joint Aeolus Tropical Atlantic Campaign (JATAC) and the ASKOS experiment, using a combination of ground-based PollyXT and Doppler lidars, satellite lidar data from Cloud-Aerosol Lidar and Infrared Pathfinder Satellite Observations (CALIPSO), radiosondes, and the model outputs of the Integrated Forecasting System (IFS) of the European Centre for Medium-Range Weather Forecasts (ECMWF). The comparison of CALIPSO lidar results with ECMWF/IFS reanalysis for 2012-2022, revealed good agreement for BL top over open ocean regions but weaker relation over dust-affected areas of the African continent. In these regions, daytime CALIPSO retrievals typically indicate lower BL tops than ECMWF, while at night CALIPSO often detects aerosols within the residual layer, leading to higher estimates than the model. Observations in Cabo Verde highlight distinctive Marine Atmospheric Boundary Layer (MABL) characteristics, such as limited diurnal evolution, but also show the potential for BL heights to reach up to 1 km, driven by factors like strong winds that increase mechanical turbulence. Additionally, the technical and physical challenges in estimating the BL height using different datasets and methods are discussed, examining cases with different thermodynamical conditions and aerosol load that directly affect the dynamics of the BL. The findings underline the strengths and limitations of different observational and modeling approaches, and emphasizes on the importance of considering local meteorology and aerosol conditions when interpreting BL height.

# 1 Introduction

The atmospheric Boundary Layer (BL) is characterized by complex interactions between surface-driven forces and meteorological conditions, which determine its height, structure, and the degree of turbulent mixing within (Stull, 1988). BL dynamics vary considerably across different environments, presenting challenges for weather modeling and prediction, especially in transitional zones like those between deserts and oceans (Seibert et al., 2000; Li et al., 2017).

<sup>&</sup>lt;sup>1</sup>National Observatory of Athens (NOA), IAASARS, Greece

<sup>&</sup>lt;sup>2</sup>Department of Physics, National and Kapodistrian University of Athens, Greece

<sup>&</sup>lt;sup>3</sup>Finnish Meteorological Institute, Finland

<sup>&</sup>lt;sup>4</sup>Atmospheric Chemistry Research Group, Chemical Resource Beneficiation, North-West University, Potchefstroom, South Africa

<sup>&</sup>lt;sup>5</sup>Leibniz Institute for Tropospheric Research (TROPOS), Leipzig, Germany

Monitoring the BL top reliably is a challenge, particularly in heterogeneous environments where traditional observation methods may fall short. Lidar systems have proven valuable for continuous profiling of aerosol and atmospheric structures, as their high vertical resolution enables detailed monitoring of BL height (Wiegner et al., 2006; Baars et al., 2008). Yet, automatic identification of the BL top from lidar data is challenging in complex areas, because BL structures can be influenced by surface type, time of day, and atmospheric stability (Tsikoudi et al., 2022). Up to now, lidar-based BL retrievals showed very good performance on relatively predictable areas with known BL patterns, such as open land surfaces or stable atmospheric conditions (Tsaknakis et al., 2011; Seidel et al., 2012). Expanding lidar BL retrievals to more complex environments, is an ongoing challenge especially when it comes to oceanic and coastal BLs where ground-based observation sites are limited.

Over the open Atlantic, the Marine Atmospheric Boundary Layer (MABL) is typically shallow and influenced by the relatively constant sea surface temperature, while boundary layers in coastal and island regions experience terrestrial-marine interactions that increase their variability (Garratt, 1994; Wood, 2012). A limited number of studies over years have addressed the detection and analysis of MABL using lidar data, primarily due to practical and observational challenges over the ocean (e.g. Atlas et al. 1986; Flamant et al. 1997; Pena et al. 2015). Given these constraints, satellite observations can provide an important means of obtaining information in remote regions lacking in-situ and ground-based remote sensing data, while also enabling the development of global climatologies (Teixeira et al., 2025).

Although the BL is a near-surface phenomenon, several satellite measurements can indirectly infer its properties, particularly its depth and spatial or temporal variability. The Cloud-Aerosol Lidar and Infrared Pathfinder Satellite Observations (CALIPSO) mission, has been widely used to derive global BL height climatologies over ocean and land and is therefore essential for studying lower troposphere characteristics (e.g. Liu et al. 2024). Nevertheless, when interpreting satellite-derived BL characteristics, it is crucial to decode the measurements appropriately, as the definition and identification of the BL can vary depending on the chosen approach and physical parameter. The Cloud-Aerosol Lidar with Orthogonal Polarization (CALIOP) of the CALIPSO satellite, can measure, among others, backscattered light from aerosols and clouds. Hence, in this case, the top of the lowest aerosol layer often coincides with the BL top, since aerosols are typically well mixed within the BL and drop sharply above it (Li et al., 2017).

The general circulation over the tropical Atlantic is dominated by the Inter-Tropical Convergence Zone (ITCZ) and affected by the presence of the Saharan Air Layer (SAL). The SAL is a typically warm and dry air layer that frequently occurs at large scales in the tropical North Atlantic Ocean and can reside up to 5 km in altitude, often accompanied by dust aerosols (Carlson and Prospero, 1972; Dunion and Velden, 2004; Wu, 2007). The ITCZ, migrates seasonally between the northern and southern tropics, influencing rainfall and convective activity, creating conditions conducive to both the formation of clouds and the aerosol convection over the Atlantic (Zhou et al., 2020). In tandem, the SAL, comprising of hot, dry air laden with desert dust from the Sahara, moves westward across the Atlantic Ocean, especially in summer, driven by the prevailing trade winds (Prospero and Mayol-Bracero, 2013) and has consequences on the surface radiation budget (Evan et al., 2009; Yu et al., 2006). These circulation patterns are key in transporting dust from Africa to the Atlantic, affecting the radiative balance and potentially impacting cloud formation, atmospheric stability, and therefore BL behavior in the region (Sun and Zhao, 2020).

A typical characteristic of the eastern sides of the Atlantic, is that the air subsiding into the subtropical north-east Atlantic is warmer and drier than the air that has been in contact with the relatively cold ocean surface influenced by upwelling, and a strong inversion forms at the interface of the two air masses (Hanson, 1991). As such, transported desert dust from Africa introduce another layer of complexity in tropospheric dynamics and clouds activity by altering radiation budget, atmospheric stability, and moisture distribution (e.g. Marinou et al. 2021; Ansmann et al. 2017; Marsham et al. 2008). This dual effect of dust—scattering and absorbing solar radiation while in the same time serving as cloud condensation and ice nucleation nuclei (CCN/IN)—leads to competing influences on the BL (e.g radiative cooling can suppress turbulent mixing, yet CCN activation can lead to increased cloud cover and associated feedback on surface radiation). These processes have been observed to influence the vertical structure and stability of the BL, but their overall impact on BL dynamics is still not fully understood.

Accurately representing BL-aerosol interactions in climate and chemical transport models is crucial because these processes affect surface conditions and large-scale atmospheric circulation (Menut et al., 2009; Pérez et al., 2006; Tombrou et al., 2015, 2007). Gaps in observational data over complex environments, such as the dust-laden, desert-ocean transition zone in the Atlantic, limit the model's ability to accurately capture BL evolution and aerosol influences (Rémy et al., 2019, 2021; Kallos et al., 2007). The need for observational data to validate and refine these models is high, especially given the impacts on cloud formation, energy distribution, and surface-air interactions. Addressing these gaps through both ground-based experimental campaigns such as Joint Aeolus Tropical Atlantic Campaign (JATAC) and satellite sensors such as space Lidars can significantly enhance understanding and modeling of BL processes in regions of critical climatic importance. In addition to investigating BL-aerosol interactions, this study aims to improve BL top detection methods in diverse and complex environments. By addressing challenges inherent to automated BL detection, particularly in areas affected by aerosols and variable atmospheric conditions, this work contributes to the development of more robust methods for BL identification.

The structure of this paper is as follows: Section 2 provides an overview of the datasets and methods used, including ground-based lidar, space lidar, radiosonde data, and model outputs. Section 3 examines the BL characteristics across different environments, beginning with the Atlantic Ocean (Area 1) and the ocean-desert transition zone (Area 2), before focusing on Cabo Verde, where dust interactions with the BL are investigated. Finally, Section 4 presents the main conclusions of this study.

#### 2 Data Sources and Analysis

55

This study uses data from the ASKOS Campaign (Marinou et al., 2023), the ground-based component of the JATAC organised by the European Space Agency (ESA). The campaign was conducted at the Ocean Science Centre Mindelo (OSCM), at the island of São Vicente, Cabo Verde, during 2021-2022. In addition, CALIPSO observations and ECMWF model data are employed. The BL height is derived using the gradient method and the Wavelet Covariance Transform method on satellite and ground-based lidar data respectively.

#### 85 2.1 Datasets



The comprehensive ASKOS dataset includes active remote-sensing observations and radiosonde profiles, both essential for characterizing atmospheric dynamics in the studied region. Specifically, ground-based PollyXT lidar and Wind Doppler lidar observations are examined, together with the LIVAS (LIdar climatology of Vertical Aerosol Structure for space-based lidar simulation studies) Climate Data Record (Amiridis et al., 2015) derived from CALIPSO. Furthermore, the BL heights obtained from measurements are compared against the ERA5 reanalysis dataset, produced with the Integrated Forecasting System (IFS) of the European Centre for Medium-Range Weather Forecasts (ECMWF), at  $0.25^{\circ} \times 0.25^{\circ}$  horizontal resolution.

# 2.1.1 Groundbased Lidars

The ground-based PollyXT Raman Lidar (Engelmann et al., 2016), consists of a compact, pulsed Nd:YAG laser, emitting at 355, 532, and 1064 nm at a 20 Hz repetition rate, with the laser beam pointed into the atmosphere at an off-zenith angle of 5°. The backscattered signal is collected by a Newtonian telescope with a 0.9m focal length, acquiring profiles with a vertical resolution of 7.5 m, and a temporal resolution of 30 s. The system was operated by the Leibniz Institute for Tropospheric Research (TROPOS) during the ASKOS Campaign, providing data coverage for the entire campaign. Figure 1 presents some indicative PollyXT measurements. Specifically, the attenuated backscatter coefficient of the 1064 nm channel (Att BSC, Fig. 1-left) is examined to derive the BL top, and the volume linear depolarization ratio (VLDR, Fig. 1-right) is complementary investigated to infer the aerosol shape. The white points in the attenuated backscatter indicate the presence of clouds and were not included in the BL analysis.

Additionally, complementary data from a Halo Photonics Stream Line scanning Doppler lidar were used to examine the horizontal wind speed and direction, as well as the vertical wind component. This lidar is a 1.5  $\mu$ m pulsed Doppler lidar with a heterodyne detector (Pearson et al., 2009). The Doppler lidar has a range resolution of 48 m and measures the attenuated aerosol backscatter and Doppler velocity along the beam direction. Horizontal wind profiles were retrieved from a velocity azimuth display (VAD) scan with 12 azimuthal angles at 60° elevation angle every 15 minutes. Otherwise, the Doppler lidar operated in vertical stare mode, retrieving vertical wind profile time series.

The Doppler lidar data was post-processed according to Vakkari et al. (2019) and a signal-to-noise ratio (SNR) threshold of 0.005 was applied to the vertically-pointing measurements. Turbulent kinetic energy (TKE) dissipation rate profiles were calculated from the vertically-pointing data using the method by O'Connor et al. (2010). Instrumental noise was calculated from signal-to-noise ratio according to Pearson et al. (2009) and subtracted from the vertical wind variance time series before the TKE dissipation rate calculation. To estimate mixed layer height (MLH) from the TKE dissipation rate profiles a threshold of  $10^{-4}m^2s^{-3}$  was applied, similar to previous studies (e.g. Vakkari et al., 2015).

### 2.1.2 Space lidar: CALIPSO-CALIOP

Towards investigating the dynamics of the BL over the Atlantic Ocean and parts of West Africa, observations of the Cloud–Aerosol Lidar with Orthogonal Polarization (CALIOP; Hunt et al. 2009), the primary instrument on board the joint National Aeronau-

**Figure 1.** Ground-based PollyXT Lidar at Mindelo (16.87°N, 24.99°W), Cabo Verde, on the 10th of September, 2021, depicting the attenuated backscatter coefficient (Att Bsc) at 1064 nm (left), and volume linear depolarization ratio (VLDR) at 532 nm (right).

tics and Space Administration (NASA) and Centre National D'Études Spatiales (CNES) Cloud-Aerosol Lidar and Infrared Pathfinder Satellite Observation (CALIPSO) mission (Winker et al., 2010), are extensively used. CALIOP provided as integrated component of the Afternoon-Train constellation of polar-orbit sun-synchronous satellites (Stephens et al., 2018), profiles of aerosols and clouds along the CALIPSO orbit-path between June 2006 and August 2023. In the framework of the study, CALIOP Level 2 (L2) Version 4 (V4) aerosol profiles (APro) of backscatter coefficient at 532 nm and particulate depolarization ratio at 532 nm are used, provided at uniform 5 km horizontal resolution and 60 m vertical resolution for the altitudinal range between -0.5 and 20.2 km above mean sea level (a.m.s.l.), for the domain encompassing the broader North Atlantic Ocean - Western Saharan Desert. Prior implementation of CALIOP optical products, rigorous quality assurance procedures are applied (Marinou et al., 2017; Proestakis et al., 2024), following also the quality controls adopted towards the generalization of the official CALIPSO Level 3 (L3) products (Winker et al., 2013; Tackett et al., 2018). Towards this objective, the most aggressive quality control procedure applied in the framework of the study is the cloud-free condition, removing the entire L2 profiles when detected atmospheric layers (Vaughan et al., 2009) along the CALIPSO orbit-path are classified as clouds in the feature-type classification algorithm (Liu et al., 2009; Zeng et al., 2019). Figure 2 provides an indicative example of the considered CALIOP observations and products, and more specifically the Feature Type (Fig.2 top left) product and the profiles of particulate depolarization ratio at 532 nm (Fig.2 top right), total backscatter coefficient at 532 nm (Fig.2 bottom left), and quality-assured total backscatter coefficient at 532 nm (Fig. 2 bottom right), along the CALIPSO overpass on the  $10^{th}$ of September 2021. For this analysis, CALIPSO data from September 2012–2022 were employed.



Figure 2. CALIPSO nighttime overpass in the ESA-ASKOS campaign region of interest in the proximity of Cabo Verde on the  $10^{th}$  of September, 2021, depicting the Feature Type (top left), particulate depolarization ratio at 532 nm (top right), total backscatter coefficient at 532 nm (bottom left), and the quality-assured total backscatter coefficient at 532 nm (bottom right).

#### 2.1.3 Radiosondes and models




Radiosonde profiles were analyzed to examine the dynamic structure of the lower troposphere and to evaluate the remote sensing measurements conducted during the intensive phase of ASKOS Campaign (June and September 2022). The GRAW DFM-09 radiosondes were launched to provide real-time, high-resolution measurements of temperature, humidity, and wind, which are essential for identifying the BL characteristics, such as height, stability, and thermodynamic properties. The sensors were equipped with a GPS receiver and transmit data via a radiofrequency link to the ground station.

The measurements-derived BL height results are compared to values obtained from the ERA5 Reanalysis dataset, produced by the ECMWF/IFS. The ERA5 data, available at a horizontal resolution of  $0.25^{\circ} \times 0.25^{\circ}$  with 137 vertical levels (Vogelezang and Holtslag, 1996), offers a consistent representation of atmospheric conditions. The BL height in the ERA5 reanalysis dataset, cannot be explicitly resolved, as turbulent processes occur on scales smaller than the model grid. Instead, it is diagnosed from boundary-layer theory based on the critical Richardson number ( $Ri_c$ ), and the parametrization of the mixed layer in the model uses a BL height from an entraining parcel model. Though, in order to get a continuous field, also in neutral and stable situations, a bulk Richardson method is used as a diagnostic, independent of the turbulence, parametrization. This method follows the conclusions of the study by Seidel et al. (2012), who showed that this algorithm is suitable for both convective and stable boundary layers, identifying a nonnegative height in all cases, and is not strongly dependent on the sounding vertical resolution. Several approximations are applied to the original algorithm—such as ignoring surface friction and setting winds near the surface (2m) to zero for radiosonde data—so that the bulk Richardson number can be consistently used to define the

BL height as the lowest level where it reaches the critical value  $(Ri_c)$  of 0.25. The detailed description can be found at ECMWF (2017), Chapter 3.

Additionally, Hybrid Single-Particle Lagrangian Integrated Trajectory (HYSPLIT) is also considered to analyze the backward trajectories of air masses arriving at the site of the ASKOS Campaign, Mindelo, Cabo Verde. This model estimates the tracking of air parcels over time, providing valuable information about the origins of the air parcels and their potential interactions with dust and other atmospheric constituents (Rolph et al., 2017). By identifying these pathways, a clearer understanding of the sources and transport mechanisms of the the atmospheric conditions at Cabo Verde can be established.

## 2.2 Boundary Layer top retrieval Methods




As mentioned above, only cloud-free lidar profiles are used in this study. The CALIPSO backscatter coefficient profiles at 532nm (Fig. 2, bottom right) were horizontally averaged over  $\pm 100$ m along the satellite trajectory around each point of interest. For instance, if the satellite crosses latitude  $16.87^{\circ}$ N, all available profiles within 200m of the trajectory are averaged. This approach improves the signal-to-noise ratio (SNR) of the measurements and facilitates comparison with coarser-resolution datasets, such as model reanalysis outputs. The BL height is then retrieved from the CALIPSO profiles using the gradient method (Li et al., 2021).

The ground-based PollyXT lidar backscatter profiles at 1064nm (Fig. 1, left) are averaged in time, since the instrument is stationary. For this study, profiles were averaged over  $\pm 15$ min or  $\pm 30$ min around the time of interest, depending on the scene homogeneity. The BL top is retrieved using the Haar Wavelet Covariance Transform (WCT) method (Brooks, 2003). The dilation factor  $\alpha$ , which defines the effective window size of the wavelet, was empirically set to 100m, corresponding to approximately 13 vertical bins, given the PollyXT vertical resolution of 7.5m. In most cases this choice provided a good balance between sensitivity to sharp gradients and noise reduction, although in some situations the dilation factor was adjusted to better capture the layering. The Haar integration is performed symmetrically above and below each altitude level over the chosen dillation window, with integration limits of  $\pm \alpha/2$  around the center altitude.

For the radiosonde data, layer detection is performed using the gradient method applied to virtual potential temperature  $(\theta_V)$  and relative humidity (RH) (Seidel et al., 2010). The  $\theta_V$ , which accounts for moisture effects on air density, provides a reliable representation of buoyancy and atmospheric stability, while RH typically exhibits a sharp gradient near the BL top in the persistently humid environment of São Vicente Island. In some cases, particularly under high aerosol loading, identification of the BL top also requires visual inspection to accurately locate the inversion cap.

Figure 3 shows profiles of the backscatter coefficient at 1064 nm from ground-based PollyXT (left), of the RH from radiosonde (middle) and backscatter coefficient at 532 nm from CALIPSO satellite lidar (right) from the  $23^{rd}$  of September 2022, around 19:30 UTC. The dotted grey lines represent the method applied for detecting BL top, namely WCT method for PollyXT Lidar and Gradient method for the rest two. A local maximum of the wavelet profile for WCT method, and a local minimum of the gradient for the gradient method, represent steep reduction in the investigated variable (orange dashed lines).

Several significant challenges arise when identifying the BL top with lidars (both ground-based and satellite), particularly in aerosol-complex environments. For a satellite-based lidar like CALIOP, the signal can become highly attenuated as it ap-

Figure 3.  $23^{rd}$  September 2022, around 19:30 UTC: Profiles of atmospheric variables and their corresponding detection methods for determining the boundary layer (BL) top. The solid blue lines represent the products of measurements, while the dotted grey lines correspond to the applied methods for BL top detection. Left: Backscatter coefficient at 1064 nm from the ground-based PollyXT lidar, Middle: Relative Humidity (RH) from radiosonde, and Right: backscatter coefficient at 532 nm from the CALIPSO satellite lidar. The grey shading in the lidar profiles represent the standard deviation resulting from the averaging. The selected BL top is highlighted by the orange dashed lines. The shading around these lines corresponds to  $\pm 50$ m.

proaches the Earth's surface, due to the existence of clouds above the BL. The weakened return signals also result from longer travel distances from the satellite platform to the earth's surface, which lead to a lowered SNR. This can compromise the reliability of detecting lower tropospheric features and lead to inaccurate identification of the BL top. To mitigate this, i) only cloud-free profiles were selected to ensure data quality, though this restriction reduces the dataset and introduces observational limitations, and ii) averaged profiles were considered to increase the SNR. Additionally, in marine environments, cumulus clouds frequently form at the BL top, which can serve as a useful, albeit indirect, marker for BL height for ground-based lidars that can detect the cloud base. Moreover, if a thin cumulus cloud is present above the BL top and allows partial laser penetration, the WCT may incorrectly identify the cloud's upper boundary as the BL top instead of the actual BL height. A similar issue occurs in the presence of dust layers, as the WCT detects reductions in the lidar signal caused by these layers. This can lead to misclassification of the dust layer boundaries as the BL top, complicating the accurate identification of the atmospheric structure. These limitations underscore the need for visual inspection to ensure accuracy in identifying the BL top in such settings, as automated methods may struggle to locate the correct layering.

# 3 Boundary Layer Characteristics in diverse environments

The characteristics of the BL during JATAC Campaign are examined across the contrasting environments depicted in Figure 4: over the Atlantic Ocean (blue rectangle - Area 1), within the ocean-desert transition zone (orange rectangle - Area 2), and

at the area of São Vicente in Cabo Verde (red circle). The Sahara Desert and the Atlantic Ocean are characterized by distinct conditions in terms of weather, aerosol concentrations, and therefore atmospheric dynamics. These variations are anticipated to influence respectively the structure and evolution of BL in the Areas of Figure 4.

The lower troposphere above the Atlantic Ocean is rich in marine aerosols, and presents relatively stable meteorological conditions, typical for open-ocean broad-scale circulations (Croft et al., 2021). In contrast, the lower troposphere over the desert is characterized by high dust aerosol concentrations, intense solar heating, and variable atmospheric stability (Giménez et al., 2010). The border region between ocean and desert introduces an interaction zone where different aerosols co-exist in big concentrations, producing unique BL characteristics due to the convergence of these differing air masses. Moreover, the existence of SAL has an impact on the on the surface radiation budget (Evan et al., 2009) and hence on the sea surface temperature (SST). Foltz and McPhaden (2008) found that Saharan dust outflows at the Tropical North Atlantic, were consistently associated with a reduction in solar radiation, with approximately 35% of SST variability attributed to dust outbreaks, while other SST cooling anomalies were linked to wind stress. The dust aerosol effect on SST depends on several factors, such as the temperature contrast between the dust layer and SST, the characteristics of the dust layer, concentration and altitude (Luo et al., 2021).

**Figure 4.** Map displaying the study areas for BL analysis: The blue rectangle (Area 1) represents the open-ocean Marine Atmospheric Boundary Layer (MABL) discussed in Section 3.1. The orange rectangle (Area 2) marks a transition zone at the ocean-desert interface, analysed in Section 3.2. The red circle is the ground-based measurements site at the Ocean Science Center Mindelo (OSCM) in Cabo Verde (3.3).

#### 3.1 Analysis of Area 1: The BL in the Atlantic Ocean


The Atlantic Ocean is characterized by dynamic weather systems and cyclonic activity, incorporating continuous exchange of heat and moisture between the sea surface and the adjacent air parcel (Schnitker, 1982). In open ocean areas such as Area 1, there is no direct interaction of the lower troposphere and the land, allowing for the development of a MABL. The MABL contains higher humidity levels and the airflow is smoother due to reduced friction from the water surface, comparing to land.

Wind and temperature profiles in the MABL are mainly influenced by sea surface temperature, oceanic currents and large-scale atmospheric circulation.





In this section, we focus on the MABL characteristics within the blue rectangle of Area 1 (Figure 4). 10 years of CALIOP data (2012–2022) are examined, using only the profiles recorded in month September. By limiting the data to one month, we aim to achieve more homogeneous conditions to better capture the prevailing environmental characteristics (e.g. relatively consistent sea surface temperatures). Figure 5-left illustrates the conceptual trajectories of the CALIPSO satellite across the study area. The analysis investigated cloud-free averaged profiles measured within 100 km around latitude 16.87° N, corresponding to the latitude of ground-based measuring site in Cabo Verde, as represented by the red points in Figure 5-left. A total of 449 profiles from nighttime and daytime CALIPSO trajectories (conceptually indicated in green and purple, respectively) are analyzed across longitudes from 60° W to 25° W. The spatial range of 100 km is suitable for capturing representative MABL characteristics in the study area because the selected profiles are cloud-free and measured over the ocean surface, maintaining generally homogeneous conditions of temperature, and humidity. For each profile, the derivative of the backscatter-coefficient profile at 532 nm is calculated (as in Fig. 3-right) and the minima are constrained at the lower 3 km.

**Figure 5.** Left: Conceptual illustration of the trajectories of the CALIPSO satellite across the study area. Right: Comparison of BL top derived from CALIPSO (blue points) and ECMWF (orange points) for 10 years (2012-2022) in Area 1.

The results of the MABL analysis from the space lidar data are compared with BL heights derived from the ECMWF dataset. To account for longitudinal time differences, each profile's measurement time is converted to local time based on its longitude. For each lidar profile, a temporally and spatially matched ECMWF point at the same local time is selected for direct comparison. The findings are presented in Figure 5-right. The blue circles display the MABL top heights derived from CALIPSO profiles, averaged hourly in local time. The orange points represent the corresponding hourly-averaged BL top heights from ECMWF. The data points are clustered within the 00:00–04:00 and 12:00–16:00 local time windows, because they correspond to CALIPSO's nighttime and daytime overpasses in the studied region for the month of September. The BL top in Area 1 under cloud-free conditions consistently ranges between 600 and 800 meters above sea level in both datasets.

There is a strong agreement in the mean BL heights between the two datasets, each exhibiting uncertainties of approximately 20%, indicating that both provide comparable estimates of the boundary layer top. This agreement suggests that CALIPSO and ECMWF are consistent in representing the overall distribution of BL heights; however, as discussed in Appendix A1, their agreement at the level of individual profiles remains limited.

While uncertainties associated with BL retrievals and time averaging may broaden the range of 600-800 m for BL top, these results are consistent with the expected behavior of the MABL, which typically exhibits limited diurnal variation. The time-averaging uncertainties shown in the figure arise from the methods used to capture the BL in the two datasets. For CALIOP profiles, lidar-based retrievals inherently carry significant uncertainty and sensitivity due to measurement noise. Here, the BL top is derived using the gradient method and aerosol layers as discussed in section 2; however, this method can occasionally detect layers that do not correspond to the actual PBL, introducing additional variability. In contrast, the model provides an averaged representation over a relatively large grid (0.25°, or approximately 27.8 km around 16°N), which may introduce variability but is less sensitive to small-scale fluctuations compared to CALIOP. Consequently, the model typically exhibits slightly lower standard deviations.

## 3.2 Analysis of Area 2: The BL in the Ocean-Desert Transition Zone






Area 2, highlighted by the orange rectangle in Figure 4, spans within longitudes of 35°W-0°: from the eastern Atlantic Ocean to the Western Africa, including the region around Cabo Verde. This area lies at the interface of two significantly different environments, as land and water interact differently with solar radiation due to their distinct heat capacities and reflective properties. On the West Africa land side, the lower troposphere directly interacts with the continental surface and the air is enriched with desert dust aerosols originating from the Sahara, where high temperatures, dry conditions, and strong winds are dominant. In contrast, the Eastern Atlantic Ocean side is predominantly influenced by marine aerosols within the lower troposphere, reflecting the ocean's stable, moisture-laden environment. In terms of heat capacity, land absorbs and releases heat quickly, leading to larger temperature fluctuations, while water absorbs energy more gradually, storing and slowly releasing it. These sharp contrasts in meteorological conditions and aerosol composition across the Area 2, are expected to have a notable impact on the BL structure.

For this analysis, similarly to section 3.1, cloud-free averaged profiles were selected from the CALIPSO satellite lidar for September 2012-2022 to derive the BL top and are compared with the corresponding ECMWF data. Figure 6 presents the BL top results obtained from CALIPSO lidar measurements (blue points), and from the corresponding ECMWF points (orange) along the cross-section at latitude 16.87° N (the latitude of the Mindelo observatory). The CALIPSO trajectories are divided into daytime (Fig. 6-left) and nighttime (Fig. 6-right) intervals after converting to local time, to highlight the distinct patterns of BL during different phases of the diurnal cycle.

In the daytime plot (Figure 6-left), the two datasets show better agreement over the ocean compared to over land. Over land, the variability increases significantly for both CALIPSO and ECMWF, sometimes reaching up to 40% (e.g., at  $lon = -8^{\circ}$ ), particularly for the ECMWF dataset. This increased variability can be attributed to the diurnal evolution of the boundary layer: the data include all BL tops from 06:00 to 18:00 local time. Since the boundary layer over land grows and decays throughout

**Figure 6.** BL height along the latitude of 16.84°N for September 2012–2022 (Area 2), derived from CALIPSO lidar (blue points) and ECMWF model data (orange points). CALIPSO trajectories were collocated with the nearest ECMWF model grid, and data were averaged over 2° longitudinal intervals. The error bars represent the variability in the BL height. The brown shaded region represents the topography of West Africa, indicating landmass and orographic features influence on the BL structure (sourced from Google Earth). The left figure illustrates results from daytime and the right illustrates nighttime CALIPSO trajectories.

these hours, typical for continental and desert areas (Garcia-Carreras et al., 2015), averaging over this period naturally results in large standard deviations. A similar behavior is observed in the CALIPSO retrievals, which also show substantial variability above land. It is also worth noting that CALIPSO tends to detect lower BL tops than ECMWF over land. This difference likely arises from the way to define the BL top: ECMWF relies on thermodynamic criteria, while CALIPSO identifies a decrease in aerosol concentration. Consequently, aerosols detected by CALIPSO are mostly confined within the mixed layer (Liu et al., 2018), whereas ECMWF's BL height may include the residual layer or even the entrainment zone above it.




In the nighttime plot(Figure 6-right), the retrieved BL tops are as expected significantly lower over land for both datasets. Over the ocean, the agreement between ECMWF and CALIPSO remains good. Over land, however, a different pattern emerges: the ECMWF dataset shows little variability but reports lower BL heights than CALIPSO, particularly further inland (lon > -10°). This again can be explained by the use of thermodynamic criteria to identify the BL top in ECMWF. In contrast, CALIPSO often detects aerosols residing in the residual layer or within the stable nocturnal boundary layer, resulting in systematically higher BL than ECMWF. An additional factor to consider is the quality of the CALIPSO nighttime profiles. The CALIOP instrument has different signal-to-noise characteristics during day and night: while solar background noise degrades daytime profiles, nighttime profiles suffer from lower photon count rates, which makes them noisier, especially over land (Hunt et al., 2009). This effect is consistent with our findings in Appendix A2, where the correlation between ECMWF and CALIPSO is low. The results suggest that the under-representation of aerosols in the ECMWF model (Morcrette et al., 2008; Bozzo et al.,

2020; Rémy et al., 2024) may also contribute to the observed differences. This effect is relevant both during daytime over land, where dust reduces incoming radiation to the surface and thus influences BL evolution, and during nighttime, when aerosols become trapped in the residual layer and can be detected by CALIPSO.

Overall, the two datasets show generally good agreement over the ocean, where both daytime and nighttime results are consistent. This aligns with the findings from section 3.1 (Area 1). The agreement is also stronger during the daytime compared to the nighttime, reflecting the limitations of the satellite nighttime measurements. Over land, however, discrepancies emerge due to the strong diurnal cycle and the different methodologies used to define the BL top.

#### 3.3 Focusing on Cabo Verde and JATAC/ASKOS





Cabo Verde is an archipelago in the eastern tropical Atlantic, with distinctive BL dynamics shaped by both the insular geography and the influence of surrounding mountains on airflow patterns. Specifically, the highest point is the Monte Verde (744m) on the eastern side, but there are also Caixa (535m) and Madeiral (680m) on the southern part, as well as Monte Cara (490 m) on the western part. Another geographical characteristic, is that Cabo Verde is situated directly in the path of frequent Saharan dust transport, so the region is often impacted by large dust plumes originating from the African continent and crossing over the islands. These dust events vary significantly in intensity, sometimes accumulating right above the BL or penetrating into it, while at other times showing minimal impact due to lower dust loads.

The islands of Cabo Verde, are located nearly 1000 km from the West African coast. The region of São Vicente spans approximately  $227 \, km^2$ , while the neighbouring (northern) island of Santo Antão covers around  $785 \, km^2$ , creating an interface where land and sea effects influence local atmospheric conditions. The origins of air drawn in to the trade winds arriving at Cape Verde are diverse depending on the season; from North America, the Atlantic, Arctic, European and African regions. During autumn, Cape Verde is situated in the direct transport pathway of easterly dust from Africa to the North Atlantic (Carpenter et al., 2010). These sea-air temperature contrasts, rough land surfaces, and fluctuating humidity contribute to a dynamic environment that reflects both marine and coastal BL characteristics.

To provide an illustrative comparison of BL results above Cabo Verde, we examine available data from radiosondes, ground-based PollyXT and Halo Lidars, CALIPSO, and ECMWF during the intensive observation periods of ASKOS (September 2021 and 2022). For this analysis, CALIPSO trajectories passing over the point of ground-based observations(16.87°N, 24.99°W) within a 300 km radius were carefully selected (Fig. 7, left). In Figure 7-right, the x-axis represents the BL top retrieved from CALIPSO ECMWF. The blue circles correspond to BL heights from ECMWF output, the red rectangles represent BL heights retrieved from the PollyXT Lidar and the black hexagons represent MLH retrieved from the Halo Lidar. The PollyXT and Halo points are fewer because the instruments were not operational during several overpasses. Additionally, only three radiosonde profiles were collocated with CALIPSO overpasses during these periods, which limits the statistical robustness of the comparison. To mitigate that, the comparison of all available radiosondes with the collocated PollyXT BL results is presented in Appendix A3. Nevertheless, they are included as examples of complementary in-situ measurements for the remote sensing datasets.

The black dashed line indicates the 1:1 line (y=x), representing perfect agreement between CALIPSO and the other datasets. The grey shaded area illustrates a  $\pm 20\%$  error margin, while the cyan shaded region corresponds to a  $\pm 100$  m error margin, providing a way to assess deviations from perfect correlation and evaluate whether the data points lie within an acceptable error range. Despite the limited sample size, we note that 77% of the PollyXT cases, 50% of the ECMWF cases, and 30% of the Halo lidar cases fall within the  $\pm 20\%$  range, suggesting that CALIPSO captures the general variability of BL height reasonably well when compared with independent datasets.



Figure 7. Left: Map showing CALIPSO trajectories (black dashed lines) passing over the ground-based observations site (red point:  $16.87^{\circ}$ N,  $24.99^{\circ}$ W) within a 300 km radius (red circle). Right: BL top retrieved from ECMWF (blue points), PollyXT Lidar (red rectangles), Halo Lidar (black hexagons) and Radiosondes (green stars) plotted against the corresponding BL heights from CALIPSO (x-axis). The black dashed line represents the 1:1 line (y = x), indicating perfect agreement. The gray shaded area denotes a  $\pm 20\%$  error margin, while the cyan shaded region corresponds to a  $\pm 100$  m error margin. The correlation lines are given as follows: i) CALIPSO-ECMWF y=0.66x+0.22 (blue line), ii) CALIPSO-PollyXT y=0.63x+0.11 (red line), iii) CALIPSO-Halo y=0.32x+0.32 (black solid line).

The correlation coefficient for PollyXT (red) and ECMWF (blue) lines, are r=0.69 and r=0.75 respectively, indicating that CALIPSO data present a rather satisfactory agreement with the model and the ground-based lidar. However, given their small positive intercepts (0.22 and 0.11), these datasets tend to estimate slightly lower BL compared to CALIPSO, even when their trends are generally aligned. The Halo lidar results, with the lowest correlation coefficient (r=0.37), show the weakest correlation with CALIPSO and the fit is inconclusive. The collocated cases may be limited, but suggest that CALIPSO generally captures the same variability in BL height as ECMWF and PollyXT, although with some systematic differences. The inconsistencies between Halo and CALIPSO BL results, reflect methodological differences, since Halo estimates the MLH from turbulence parameters while CALIPSO relies on gradient-based detection of layering. Similarly, ECMWF uses a thermodynamic approach (according to ECMWF, ch. 3), which may also contribute to discrepancies. The best agreement is found between the two aerosol lidars, highlighting that the choice of parameter used to define the BL height is critical for the as-

sessment. CALIPSO and Polly use aerosols as tracers, identifying the BL top from the sharp reduction in aerosol load at the transition to the free troposphere, whereas the Halo determines the BL height from turbulence, calculated through vertical velocity variance.

## 3.3.1 Dust Layer above the Marine BL




Figure 8 shows HYSPLIT backward trajectories overlaid on the SST data from the ECMWF model. The trajectories trace the air masses 48 hours prior to September 12, 2022, at 16:00 (close to the radiosonde launch time), with altitudes at 500 m, 1000 m, and 2600 m. The air at 500 m and 1000 m (black dashed and grey) in Cabo Verde originate over cooler SSTs near the African shoreline (blue dashed-dotted), while the air from higher levels (2600 m-green) comes from the African continent, likely transporting desert dust.

**Figure 8.** HYSPLIT backward trajectories depict air masses arriving in Mindelo, Cabo Verde, at altitudes of 500 m (black dashed line), 1000 m (grey solid line), and 2600 m (blue dashed-dotted line), 48 hours prior to 16:00 UTC on 12 September 2022, overlaid on ECMWF sea surface temperature (SST) data.

As previously discussed, it is common to observe dust layers transported from Africa to Cabo Verde, creating a distinct layering effect (Carpenter et al., 2010). At lower levels, the marine air mass is in direct contact with the sea surface, while a dust layer lies above it (Tsikoudi et al., 2023). These two layers differ significantly in stability and aerosol composition, resulting in a stratified profile where the dust layer rests on top of the BL. Figure 9a, illustrates the Volume Depolarization Ratio (VLDR) of the 532nm channel from the PollyXT lidar, combined with radiosonde profiles. The greenish colour in the colorbar represents non-spherical aerosols, with depolarization values around 20%, indicative of dust particles. The PollyXT lidar data are plotted for a 30-minute period surrounding the radiosonde launch time (16:19 UTC), ensuring a close temporal match between the two datasets. The relative humidity (blue) and virtual potential temperature (red) profiles from the radiosonde reveal a pronounced

Figure 9. a) Radiosonde profiles for relative humidity (blue line), virtual potential temperature  $\theta_V$  (red line), wind speed (magenta line), and wind direction (black stars) are plotted over the Volume Depolarization Ratio at 532 nm ( $VLDR_{532}$ ) from the PollyXT lidar, within 30 minutes around the launch time at 16:19 UTC on 12 September 2022 (16:04-16:34 UTC). b) Profile of attenuated backscatter coefficient at 1064 nm (black line,  $\beta_{1064}$ ), averaged over the same 30-minute window, with the grey line indicating the WCT, the dashed red line marking the BL height from PollyXT ( $BL_{PollyXT}$ ) at 650 m, the dotted orange line and the dotted blue line marking the BL from ECMWF ( $BL_{ECMWF}$ ) and radiosonde ( $BL_{RS}$ ) at 760 m and 1.1 km respectively. c) Halo Wind Doppler Lidar Turbulent Kindetic Energy (TKE) dissipation rate for the same 30-minute period. The black hexagons represent the Mixing Layer Height (MLH).

inversion near 1 km, aligning well with the stratified layers observed in the depolarization data from the lidar. This inversion acts as a cap, limiting vertical mixing and promoting layer stratification. Additionally, a subtle inversion is present around 500 m in the humidity profile, which may suggest another layered structure. The wind direction (black) remains predominantly northeasterly, with a marked increase in wind speed between 1 and 1.3 km. The BL top, could be signified along the strong humidity inversion, around 1 km. Up to this range,  $\theta_V$  is nearly constant with height, where thermal and mechanical eddies enhance turbulent mixing and redistribute heat and moisture. Higher than 1km,  $\theta_V$  increases suggesting stable stratification.

Figure 9b presents the attenuated backscatter coefficient ( $\bar{\beta}_{1064}$ ) profile at 1064 nm from the PollyXT lidar (black line). The profile is averaged over a 30-minute period around the radiosonde launch time (16:24–16:34 UTC). The grey line represents the WCT method, with its maximum indicating a layer top at 650 m (red dashed line). For comparison, the ECMWF BL top at the radiosonde launch time is shown as a dotted orange line at 760 m and the radiosonde BL top as a dotted blue line at 1.1km. The TKE dissipation rate from the Halo Wind Lidar (9c) shows larger values below approximately 520 m, aligning with the identified MLH (black hexagons).

Upon assessing all the BL results together, we find that the two lidars are in good agreement, consistently capturing the well-mixed aerosol layer. In contrast, the radiosonde indicates the strongest inversion at around 1 km, which is relatively high for a BL in this region and differs significantly from the lidar results. As discussed by Brooks et al. (2017), an apparently well-mixed potential temperature profile may extend into a residual layer where turbulent mixing is no longer active, leading to an overestimation of the actual BL height. Moreover, while lidar detects the top of the aerosol mixing layer, radiosondes diagnose stability changes that may reflect remnants of earlier mixing. Therefore, these differences between lidar- and radiosondederived boundary layer heights can be expected, particularly under conditions of weak turbulence or decoupled layers.

## 375 3.3.2 Desert Dust within the Marine BL






According to the HYSPLIT trajectories in Fig. 10, the air masses arriving over Mindelo at 1000 m and 2000 m altitudes originate from inland Africa, while the lower-level air mass, reaching 500 m, follows a path from the northwest coastline. This again indicates an influx of air masses with distinct characteristics, where the higher layers likely carry Saharan dust, in line with the VLDR measurements of PollyXT Lidar. Additionally, Aerosol Optical Depth (AOD) measurements from the Aerosol Robotic Network (AERONET) for this day report values around 0.6 at 500 nm (data not shown), further supporting the presence of significant dust transport.

Turbulence at the top of a daytime BL, driven by surface heating and convection, can lead to the entrainment of dust particles from an elevated layer above into the BL (Marsham et al., 2008). In these situations, the dust particles become integrated into the marine and coastal air masses, impacting aerosol concentrations and BL dynamics. In Figure 11a, the values of VLDR inside the BL are close to 20%, indicating the existence of dust particles in the MABL, mixed with marine particles. The radiosonde profiles of virtual potential temperature and relative humidity reveal weaker inversions than those observed in Section 3.3.1, with a notable inversion around 500 m, which indicates the approximate BL top in this case, since the  $\theta_V$  increases and RH begins to decrease at this point. The weakened inversions also suggest that the BL may be more susceptible to vertical mixing, facilitating dust intrusion from higher altitudes into the BL. On this particular day, the wind speed profile (magenta line) shows

**Figure 10.** Same as Figure 8 for 23 September 2022. The backward trajectories are calculated at altitudes of 400 m (black dashed line), 1000 m (grey solid line), and 2000 m (blue dashed-dotted line), 48 hours prior to 19:00 UTC on 23 September 2022.

milder conditions, reaching speeds up to 10 m/s (~5 on the Beaufort scale). The direction of the wind is northern (black stars) relatively to the previous case.

The WCT method (grey line) applied to the averaged  $\bar{\beta}_{1064}$  profile (Fig. 11b) identifies the BL top at 560m (red dashed line), that correspond to the most pronounced feature below 1.5km. It is worth noting that none of the WCT maxima is particularly dominant, due to the widespread aerosol load within the first 3km. This highlights a limitation of the method when the lidar signal is influenced by overlying features, such as elevated aerosol layers or thin cirrus clouds (Brooks, 2003). In such cases, cross-checking the results with independent measurements is essential. According to the wind lidar, turbulent motions, as inferred from the TKE dissipation rate, extend up to 600m (Fig. 11c), while the ECMWF boundary layer top for the same time is located at 720m (dashed orange line). The two ground-based lidars and the radiosonde show good agreement, supporting a BL top around 500–600m. The ECMWF BL top is approximately 200m higher than the other estimates, whereas in the previous case its deviation from the lidar results was smaller. In both cases, however, these differences are within the expected variability, given the model's coarse horizontal resolution of 0.25° ( 27km).

## 4 Conclusions




This study highlights the critical importance of understanding the BL in the Atlantic, to better characterize the complex interactions between the ocean and the atmosphere, particularly in the presence of transported Saharan dust. These interactions govern fundamental processes such as evaporation, sea surface temperature variability, and cloud formation, all of which have significant implications for climate modelling and marine ecosystem productivity due to dust nutrient deposition.

Figure 11. Same as Figure 9 for 23 September 2022. a) The radiosonde launch time at 19:38 UTC on 23 September 2022. b) PollyXT Lidar BL height at 560 m (dashed red line,  $BL_{PollyXT}$ ), radiosonde BL height at 520m (dotted blue line,  $BL_{RS}$ ), and ECMWF BL height at 720 m (dotted orange line,  $BL_{ECMWF}$ ). c) Halo Wind Doppler Lidar TKE dissipation rate for the same time period as PollyXT, with the black hexagons representing the MLH.

Our findings demonstrate that, based on September data of 10 years (2012-2022) of CALIPSO measurements over the open Atlantic (Area 1), the BL height ranges from 600 m to 800 m above mean sea level for both daytime and nighttime trajectories, with only cloud-free profiles considered. These results are in strong agreement with ECMWF estimates, with both datasets exhibiting uncertainties of about 20%. While CALIPSO retrievals are more sensitive to noise and local aerosol variability, and ECMWF represents a coarser spatial average, their mean values align well, suggesting that both approaches provide consistent estimates of the MABL height in the open Atlantic.



The comparison of CALIPSO and ECMWF data over Area 2, highlights the contrasting behavior of the BL over land and ocean. Over the ocean, both datasets show consistent BL heights during day and night, in line with the results from Area 1. Over land, however, larger discrepancies emerge, particularly during daytime when the strong diurnal cycle drives large variability, and at night when CALIPSO often detects aerosols in the residual layer while ECMWF applies thermodynamic criteria. These differences are further influenced by the under-representation of aerosols in ECMWF (Morcrette et al., 2008; Bozzo et al.,

2020; Rémy et al., 2024) and the inherent limitations of CALIPSO nighttime retrievals. Overall, while the two datasets provide a consistent picture of the MABL, care must be taken in interpreting BL results in land, where methodological and physical factors can lead to significant divergence of the satellite and the model.

In Cabo Verde, collocated data from CALIPSO, PollyXT, Halo Lidar and radiosondes were analyzed for September 2021–2022. The results show that CALIPSO is able to capture the general variability of the boundary layer when compared with independent datasets, particularly with PollyXT and ECMWF. The differences observed across the instruments largely reflect the distinct definitions and retrieval methods used to estimate the BL top, emphasizing that no single dataset provides a complete picture on its own. To further investigate the situation in Cabo Verde, two cases with distinct aerosol loads and thermodynamic conditions were examined. The first case (12 September 2022) is characterized by dust aerosols primarily above the capping layer, while in the second case (23 September 2022), the dust aerosols have penetrated the BL. The two ground-based lidars show good agreement in both cases, while the radiosonde BL top is found to be much higher in the first case. Furthermore, when dust intrudes into the BL, radiosonde inversions may be very weak, making it hard to determine the BL top with confidence. Similarly the lidar signal can be influenced by multiple layers, and methods such as the WCT may not yield a clearly dominant maximum, highlighting the need for careful interpretation and cross-validation, as it is challenging to automate the BL identification process.

It is important to note that differences between the BL heights derived from different instruments/model and methods do not necessarily imply that one is correct and the other is wrong. Rather, they often reflect the fact that each technique responds to a different physical aspect of the boundary layer. For instance, in ERA5, the BL top is not explicitly resolved but diagnosed from boundary-layer theory using a critical  $Ri_c$ , representing the depth of active turbulent mixing. Radiosonde-derived heights are typically based on thermodynamic structure, identifying the strongest temperature or humidity inversion, which may correspond to a residual layer rather than the actively mixed layer. Lidar measurements, in contrast, detect gradients in aerosol backscatter, which trace the extent of aerosol mixing but may remain unchanged even after turbulence ceases. An apparently well-mixed potential temperature profile may thus extend well above the dynamically defined boundary layer (Brooks et al., 2017). Therefore, the discrepancies observed between model, lidars, and radiosonde estimates likely arise because these approaches describe related but not identical layers within the lower atmosphere.

The variability of the atmospheric conditions in the studied region is driven by the combined influence of marine and dust aerosols together with the complex sea—land interactions. Hence, the height detected for the BL top needs careful treatment and the interpretation is highly dependent on the definition and methods used. Lidars typically identify the top of aerosol layers, which may coincide with the BL top, but it is crucial to combine with multiple instruments and account for local characteristics and aerosol conditions for a robust estimation of the BL height. This study suggests that when these complex conditions favor less instability, desert dust from the SAL is more efficiently penetrating to the BL. This mechanism should be further examined on its importance as a facilitator of dust deposition to the ocean. Experiments such as JATAC bring the observational synergies needed to study complex BL dynamics governing dust transport.

Data availability. The ASKOS Campaign dataset is available from the ESA Atmospheric Validation Data Centre (EVDC) at https://evdc.esa.int with DOI: 10.60621/jatac.campaign.2021.2022.caboverde (Marinou et al., 2023). Visualized datasets of the ASKOS Campaign and additional information are also available at https://askos.space.noa.gr/data; The ERA5 reanalysis dataset used in this study are available from the Copernicus Climate Change Service (C3S) Climate Data Store: Hersbach et al.: ERA5 hourly data on single levels from 1940 to present, (C3S - CDS), https://doi.org/10.24381/cds.adbb2d47 (accessed on 30-MAY-2025). The livas dataset is available upon request.

# Appendix A: Statistical analysis


The following subsections present the statistical analysis of boundary layer heights from ECMWF and CALIPSO in areas 1 and 2, that are discussed in sections 3.1 and 3.2 respectively.

## A1 Area 1 Statistical analysis

**Figure A1.** Distribution and intercomparison of BL heights from CALIPSO and ECMWF for Area 1 (2012–2022). Left panel: Normalized histograms and kernel density estimates (KDE) for both datasets. The shaded hatched region highlights the overlap of the two histograms (84%), indicating a strong similarity in the overall distributions. Right panel: Scatter plot of collocated BL heights, with linear regression (blue line) and statistics of the fit.

The two datasets exhibit very similar distributions, with almost identical means (Figure A1:  $734 \pm 203$  m for CALIPSO and  $735 \pm 161$  m for ECMWF). The shaded region indicates the overlap of the histograms (84%), and confirm a strong similarity in their climatological distributions. Despite the high overlap, the regression line (blue) shows a statistically significant but only moderate correlation (r = 0.50). This suggests that although CALIPSO and ECMWF are consistent in representing the general

distribution of BL heights, their agreement at the individual profile level remains limited. The high distributional overlap points to a reliable representation of the mean state by both datasets, while the relatively low correlation indicates differences in the day-to-day variability captured by the satellite retrievals and the reanalysis. These differences likely arise from the distinct BL identification techniques: CALIPSO relies on an aerosol-based approach and ECMWF on a thermodynamics-based approach.

# A2 Area 2 Statistical analysis


**Figure A2.** Distribution of BL heights from CALIPSO (blue) and ECMWF (orange) for (left) daytime and (right) nighttime conditions. Solid histograms show the frequency of occurrence, with dashed lines indicating the corresponding kernel density estimates (KDEs). The shaded areas mark the overlap between the two datasets: 72% during daytime and 64% during nighttime.

In Figure A2-left, the KDEs reveal a bimodal distribution in the daytime data. The datasets tend to capture two distinct boundary layer regimes: the taller and narrower peak around 800 m corresponds to oceanic boundary layer conditions, representing points that occur more frequently and consistently. The shorter and broader peak around 2100 m corresponds to land conditions, which occur less often and with greater variability. This interpretation is consistent with the large standard deviations (1351  $\pm$  803 m for CALIPSO and 1486  $\pm$  1095 m for ECMWF). At night (Figure A2-right), the CALIPSO boundary layer heights are more dispersed, with some indication of a secondary mode near 1100 m that likely reflects elevated BL values over land. By contrast, the ECMWF data cluster more tightly around 500 m, forming a clearer unimodal distribution.

The overall correlation between CALIPSO and ECMWF boundary layer heights is high during daytime (r = 0.89). However, when separating ocean and land points, the correlations are notably lower (r = 0.56 for ocean, r = 0.42 for land). At nighttime,

**Figure A3.** Scatter plots of BL heights from CALIPSO versus ECMWF for (left) daytime and (right) nighttime. Black points indicate land observations, fitted with the maroon line; white points indicate ocean observations, fitted with the blue line; all data together are fitted with the solid red line. Correlation coefficients (r) are indicated for each subset.

the correlation improves slightly for ocean points (r = 0.64) but remains moderate, while the overall correlation between CALIPSO and ECMWF points is poor.

#### 480 A3 Cabo Verde Radiosondes and Lidar BL results correlation



Figure A4 shows a correlation plot comparing BL heights retrieved from radiosondes and the PollyXT Lidar for all available collocated measurements (N = 40). Out of the 50 radiosondes launched during the intensive phase of the campaign, 10 cases were excluded due to cloud contamination or periods when the PollyXT Lidar was not operational. The correlation coefficient is r = 0.87, indicating a strong agreement between the two datasets. Most points fall slightly below the y = x line, suggesting that the radiosondes tend to detect slightly higher BL compared to PollyXT Lidar.

Author contributions. IT and EM conducted the analysis and drafted the manuscript; MT, EG and VA provided methodological guidance and contributed to the interpretation of the data; EM, MT and VA designed the study framework and defined the research objectives; EP, KR and VV contributed to the data curation, processing, visualization and revisions of the results; HB, RE, and AS contributed to the PollyXT data acquisition; HB and ZY assured the Pollynet data processing; EM and VA provided funding acquisition and project administration; All authors edited and reviewed the original draft, provided critical feedback and helped shape the research, analysis and manuscript.

Figure A4. Correlation between boundary layer heights retrieved from radiosondes and the PollyXT Lidar for June and September 2022 (N=40). The grey dotted line represents the 1:1 line (y=x), while the grey dashed lines indicate  $\pm 150$  m from the 1:1 line. The blue dashed line shows the linear regression fit (y=0.91+4.84) and correlation coefficient r=0.87 highlighting the strong correlation between the two datasets.

Competing interests. The authors declare that they have no competing interests.



Acknowledgements. This research has been supported by the Hellenic Foundation for Research and Innovation (H.F.R.I.) under the "3rd Call for H.F.R.I. Research Projects to support Post-Doctoral Researchers" (Project Acronym: REVEAL, Project Number: 07222). We also acknowledge the support by the PANGEA4CalVal project (Grant Agreement 101079201) funded by the European Union . Emmanouil Proestakis acknowledges support by the AXA Research Fund for postdoctoral researchers under the project entitled "Earth Observation for Air-Quality – Dust Fine-Mode (EO4AQ-DustFM)". Funding was also received from Horizon Europe programme under Grant Agreement No 101137680 via project CERTAINTY (Cloud-aERosol inTeractions & their impActs IN The earth sYstem). Parts of this research have been supported by the German Federal Ministry for Economic Affairs and Energy (BMWi) (grant no. 50EE1721C) and by the German Federal Ministry of Education and Research (BMBF) under the FONA Strategy "Research for Sustainability" (grant no. 01LK2001A). Finally, the authors would like to acknowledge the project of ASKOS (Grant agreement 4000131861/20/NL/IA) from the European Space Agency. The authors would also like to thank the anonymous referees for their reviews, which improved the paper significantly.

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
