# Peer review of "Atmospheric Boundary Layer in the Atlantic: the desert dust impact"

_EGUsphere, 2025_

## Author Comment (AC1)

AUTHORS' STATEMENT: *The authors would like to thank the referees for the time and care they devoted to review this manuscript, and for their constructive comments. We have made substantial revisions to both the analysis and the presentation of the results. We hope that our responses and the significant modifications have satisfactorily addressed the concerns raised and improved the overall quality of the manuscript. Referee comments are listed below, and our responses are provided in* blue font. *Line numbers in our responses refer to the revised version.*

**ANONYMOUS REFEREE 1**

**Summary**

The paper examines the Boundary Layer over marine and West African regions. It utilizes various instruments to compute the Planetary Boundary Layer Height (PBLH) using both space-based and ground-based measurements, along with ECMWF outputs. The study discusses the similarities and differences among the various technologies and retrieval algorithms employed. It effectively characterizes the horizontal variability of the PBLH in marine and land regions during September and compare it to ECMWF retrievals. However, questions about the methodology and robustness of some analyses are problematic.

**Major comments:**

- Why did you choose to use Relative Humidity (RH) to calculate the PBLH from sonde measurements, instead of using potential temperature profiles? Potential temperature is typically the more common variable utilized to derive the PBLH from radiosondes. RH measurements are often avoided due to their higher uncertainty, making them less reliable compared to potential temperature (see Liu and Liang, 2010).

We thank the reviewer for this valuable comment. We fully agree that the most scientifically robust and commonly used variables for determining the BL height from radiosondes are potential temperature and specific humidity (Seidel et al., 2010; Liu and Liang, 2010).

Relative humidity, is a variable that depends on both temperature and water vapor content, and thus varies strongly. In contrast, specific humidity is the mass of water vapor per unit mass of moist air and remains nearly conserved for an air parcel in the absence of condensation or evaporation. In our case, the radiosonde measurements were conducted on São Vicente Island in Cabo Verde, a tropical marine region characterized by a persistently humid environment. Due to this high humidity, we opted to examine the virtual potential temperature ($\theta_V$), which accounts for moisture effects on air density and provides a better representation of buoyancy and atmospheric stability in moist environments.

Additionally, we observed that under such humid conditions, RH exhibits a sharp gradient at the BL top, due to the moist marine layer capped by drier air aloft. While indeed RH is not typically recommended as a primary variable for examining the BL, in our case it provided a clear empirical signal of the BL top. We present two examples of radiosonde data displaying RH, specific humidity, water vapor mixing ratio, potential temperature and virtual potential temperature in Figure 1. As we can see, the inversions (orange shading) in all the profiles for each case are located at the same level for all parameters. Therefore, this consistency supports the use of these moisture-related parameters for identifying the top of the BL. This is the reason we have included RH in Figures 3, 9, and 11 of the manuscript, where it complements the interpretation based on virtual potential temperature, but also on other measurements as well ($VLDR_{532nm}, \beta_{1064nm}, TKE_{dr}$). However, our methodology section did not sufficiently explain the rationale behind the inclusion of RH in our analysis. We have revised the manuscript accordingly (lines 162-166 of the new version) to clarify this approach.

- Is there any reason why you chose to do these analyses in September? It would also be worthwhile to evaluate other months, especially when the SAL activity typically ramps up between mid-June and mid-August.

The choice of September for our analysis is primarily driven by having more homogeneous conditions

[Figure]

Figure 1: Moisture variables (green:water vapor mixing ratio, purple: specific humidity, blue:RH) and temperature variables (red: potential temperature ($\theta$), magenta: virtual potential temperature($\theta_V$)) for the Radiosonde launches of (a) 12 sep 2022, 08:53 UTC and (b) 21 Sep 2022, 20:44 UTC

to better capture the prevailing environmental characteristics (lines 211-213 of the revised manuscript), but also by the data availability and the observational strategy of the ESA JATAC campaign (see this link). While the campaign extended through 2021–2022, the intensive observation phases with ground-based measurements were limited to September 2021, June 2022, and September 2022. Radiosondes were only launched during 2022, close to the ground-based lidars. These complementary datasets are critical to our study.

Furthermore, September presents distinct thermodynamic and aerosol conditions. As shown in Figure 2, sea surface temperatures are significantly warmer in September compared to June, which can influence the vertical structure of the marine boundary layer, and generally the lower troposphere dynamics. Additionally, we observed that September typically features fewer low-level clouds over the Cabo Verde region compared to June, offering clearer and more reliable lidar retrievals (Marinou et al., 2023).

[Figure]

Figure 2: Sea Surface temperature (SST) from ECMWF ERA5 Reanalysis for (a) 15 June 2022 and (b) 15 Sep 2022.

Although SAL activity ramps up during mid-June to mid-August (source), dust presence remains strong through September, as we illustrate in Figure 3 for September 2021 (a) and 2022 (b). Having selected September as our focus month, we consistently compare data across the same month in different years to ensure comparable SST and seasonal conditions. Evaluating other months would be a valuable extension of this study and could be explored in future work, especially with EarthCARE now in operation.

[Figure]

Figure 3: Volume Depolarization Ratio at 532 nm from the ground-based PollyXT Lidar in Mindelo, Cabo Verde for (a) September 2021 and (b) September 2022. The Saharan Air Layer (SAL) is present at 1-5km during almost the entire month.

- While the slope of a linear regression and the correlation coefficient are related, they are not same. The authors argue that there is a significant correlation among the different retrieved PBLHs. However, they fail to include any correlation coefficients, basing their claims solely on the slope. Additionally, the analyses presented in Sections 3.1 and 3.2 would benefit from the inclusion of scatter plots and correlation coefficients (or the $R^2$ value from the linear regression) to better illustrate the agreement between the CALIPSO and ECMWF data. Although the mean values may be similar, the variability from day to day can differ significantly.

We thank the reviewer for pointing this out. The slope of a linear regression provides information about the scaling and potential biases between datasets, but indeed it does not capture the strength of their linear relationship. To address this, we have included the correlation coefficient (r) in the scatter plot comparing the daily CALIPSO and ECMWF BL heights (Section 3.3, figure 7 of the new version), along with the corresponding description in the text. Furthermore, we have added an Appendix with the statistical analysis from Sections 3.1 and 3.2, that presents the distribution and intercomparison of BL heights from CALIPSO and ECMWF for Areas 1 and 2. We included normalized histograms and kernel density estimates (KDE) for both datasets, as well as scatter plots with linear regression. We modified the discussion of the following sections:

[revised manuscript text omitted]

We hope that this additional information allows for a clearer assessment of both the agreement and variability between the two datasets.

- Constraining your analyses to only September limits your investigation's robustness, especially in section 3.3. In the radiosonde measurements, you only have 3 cases, which does not allow you to even make any meaningful conclusion about the relationship between the retrieved PBLH from the radiosonde and CALIPSO.

We acknowledge this concern. The number of radiosondes collocated with CALIPSO overpasses at Cabo Verde is limited, since radiosondes were launched only during the 2022 intensive observation periods and not systematically aligned with CALIPSO trajectories (see line 125 of the new version). As a result, only a few cases are available for direct comparison. Nevertheless, we consider these cases valuable as they provide rare, collocated in-situ and satellite observations of the BL. While they do not allow for robust statistical conclusions, they do offer illustrative examples that support our interpretation and highlight the challenges of obtaining collocated datasets in such remote marine locations. We have revised Section 3.3 to clarify this point and to present the comparison as case-based rather than a statistical evaluation. For example we mention that *"Additionally, only three radiosonde profiles were collocated with CALIPSO overpasses during these periods, which limits the statistical robustness of the comparison. Nevertheless,*

*they are included as examples of complementary in-situ measurements for the remote sensing datasets".* Moreover, we have added in the appendix A3 a correlation plot comparing the BL heights from all available radiosondes (N = 40) with the collocated in time PollyXT Lidar retrievals to demonstrate the good agreement between the two datasets with correlation coefficient $r = 0.87$.

- I think the paper would benefit from a richer discussion about the improvements needed in ECMWF in representing the PBLH, especially during the continental region of West Africa, where you observed the highest differences.

We have substantially revised the sections discussing the representation of the BL height in ECMWF and the results of the comparisons, with particular attention to the continental region of West Africa where the largest differences were observed. In the revised version, we have expanded both the discussion, the methodology with more information on the ECMWF BL derivation, and the of course the conclusions. We have used all available datasets, while staying within the limits of the observational capabilities. We believe that this has allowed us to address the gaps present in the previous version as thoroughly as possible. We sincerely thank the reviewer for the valuable comments, which provided the triggering for a deeper investigation and has significantly improved the quality of the manuscript.

**Minor comments:**

- Although the authors mentioned the instruments and techniques used to compute the PBLH, the description of the methodology is missing some information that will allow its reproduction. In particular, the WCT is very sensitive to the dilation factor in the Haar function, but this term is not included in the methodology section, nor are the integration limits. In addition, people typically preprocess lidar data before computing the PBLH, either by interpolating between pressure intervals or horizontally averaging to increase the SNR. I wonder if the authors made anything like this

We have revised the methodology section to ensure all details necessary for reproducing the analysis are included. In particular, we now specify the dilation factor $\alpha$ used in the Haar WCT, which was empirically set to 100m, although it is occasionally adjusted to better capture layering. We also provide more details regarding the preprocessing of the lidar profiles: CALIPSO profiles are horizontally averaged over ±100m along the trajectory, while PollyXT profiles are averaged in time over ±15min or ±30min, depending on scene homogeneity. This is illustrated ion the updated figure 3. Furthermore, we have reproduced the analysis using this updated CALIPSO averaging (±100m), and the results based on this approach are now included. We hope that these additions make the methodology fully transparent and reproducible, while noting that the analysis was performed for specific environments -marine or aerosol affected- and that areas with different characteristics may require tailored handling.

- L50: Spaceborne lidar signal not only attenuates as it approaches the surface due to the presence of clouds. The weakened return signals result from longer travel distances from the satellite platform to the earth's surface, which lead to a lowered SNR.

We believe the reviewer means line 150 instead of 50. We have modified the text as *"For a satellite-based lidar like CALIOP, the signal can become highly attenuated as it approaches the Earth's surface, due to the existence of clouds above the BL. The weakened return signals also result from longer travel distances from the satellite platform to the earth's surface, which lead to a lowered SNR. This can compromise the reliability of detecting lower tropospheric features and lead to inaccurate identification of the BL top. To mitigate this, i) only cloud- free profiles were selected to ensure data quality, though this restriction reduces the dataset and introduces observational limitations, and ii) averaged profiles were considered to increase the SNR"* (Lines 173-178 of the revised version).

- Figure 3: The authors mentioned that the PBLH was retrieved using both the WCT and the gradient methods for Lidar measurements. However, in Figure 3, the authors only showed the WCT for the PollyXT and the gradient for CALIPSO.

We apologize for the confusion, indeed this was not clearly presented. What we meant is that the WCT and gradient methods were applied to the ground-based and satellite lidar, respectively. We recognize that this may have been confusing, and we hope that the revised methodology section now clarifies this point.

- In line with the previous comment, you should also include in Figures 5 and 6.

the retrieved PBLH from the WCT for the CALIPSO data.

To clarify, we have applied the gradient method to the CALIPSO data, not the WCT. We apologize for any confusion and hope that this is now clear in the revised version. The WCT is a more complex method and it is a bit more difficult to apply to satellite data and less straightforward to automate. Consequently, the PBLH shown in Figures 5 and 6 is based on the gradient method, and including WCT for CALIPSO is not feasible for this study.

- Why did you choose to use Relative Humidity (RH) to calculate the PBLH from sonde measurements, instead of using potential temperature profiles? Potential temperature is typically the more common variable utilized to derive the PBLH from radiosondes. RH measurements are often avoided due to their higher uncertainty, making them less reliable compared to potential temperature (see Liu and Liang, 2010).

This point has been addressed in the response to the first major comment, where we explain that both virtual potential temperature and relative humidity were used to detect the BL top. We also note that there was an error in Figures 9 and 10, where the actual temperature was plotted; this has now been corrected.

---

## Author Comment (AC2)

AUTHORS' STATEMENT: *The authors would like to thank the referees for the time and care they devoted to review this manuscript, and for their constructive comments. We have made substantial revisions to both the analysis and the presentation of the results. We hope that our responses and the significant modifications have satisfactorily addressed the concerns raised and improved the overall quality of the manuscript. Referee comments are listed below, and our responses are provided in* red font. *Line numbers in our responses refer to the revised version.*

**ANONYMOUS REFEREE 2**

**Summary**

The paper describes boundary layer (BL) height characteristics over the subtropical northern Atlantic off the coast of Africa, as derived from CALIPSO observations, ECMWF/IFS reanalysis, and ground-based observations comprising of two ground-based lidars and radiosonde observations. Ten-year BL height climatological values over two regions are analyzed and intercompared using CALIPSO observations and ECMWF reanalysis. The ground-based lidars and their respective BL height retrieval algorithms are evaluated and compared against CALIPSO using data from Cabo Verde. Furthermore, two test cases over Cabo Verde are evaluated showcasing distinct interactions between the BL and the Saharan Air Layer (SAL). The first case shows stronger BL inversions and suggests clear separation between SAL and BL, whereas the second case exhibits weaker inversion and shows dust aerosols mixed throughout the BL.

I commend the authors for assembling and performing analysis of several different datasets. The topic is interesting, and the figures are engaging, although the figure fonts should be substantially enlarged. The writing is largely clear and understandable, with only sporadic improvements of style required. The paper shows potential, although in my opinion the paper it falls short on meaningfully investigating the impact of dust on the Atlantic BL. It seems to me rather showcasing a collection of measurements and datasets, with little and inconclusive analysis of their strengths and disadvantages. Here are a few specific complaints:

**Major comments:**

1. Climatological analysis of collocated CALIPSO and ECMWF (sections 3.1 and 3.2).
The results section starts with analysis of climatological values of BL height in Area 1 and 2. Over Area 1, CALIPSO and ECMWF are in general agreement, with ECMWF being slightly higher than CALIPSO. However, in Area 2, and especially over land, we see very large differences in BL heights. The authors argue, that "CALIPSO in some cases detects the mixing layer height rather than the residual layer and the entrainment zone (Liu et al, 2018)", an explanation that is vague and unsatisfactory. The large differences between datasets require more in-depth analysis. For example, how many profiles were used in each of the bins in Fig. 6? How often CALIPSO misidentifies BL height in these cases (the error bars on CALIPSO data suggest it is a systematic bias rather than occasional misidentification)? Why two over-land bins agree within the error bars, but six bins do not? For nighttime data CALIPSO is systematically higher than ECMWF. This is very interesting, but it is not mentioned in the manuscript and no explanation is provided.

We sincerely thank the reviewer for these fruitful comments. We have performed major revisions to the manuscript and repeated parts of the analysis to address these concerns.
We paid particular attention to the averaging of CALIPSO data. The large differences and high variability in CALIPSO BL heights needed careful handling. To increase SNR of Calipso data, we averaged all profiles over ±100m around the point of interest. This averaging is a common technique in satellite data and significantly reduced variability. The updated discussion in Section 3.2 reflects these changes, as well as the better consistency of the two datasets.
For reference, this averaging resulted in 432 profiles for Figure 6 and 549 profiles for Figure 5. Additionally, the methodology section has been updated, and a statistical analysis has been added in the Appendix. The revised discussion in Section 3.2 now also addresses the systematically higher CALIPSO

BL heights during nighttime, as observed by the reviewer, along with other modifications to improve clarity and interpretation: *In the daytime plot (Figure 6-left), the two datasets show better agreement over the ocean compared to over land. Over land, the variability increases significantly for both CALIPSO and ECMWF, sometimes reaching up to 40% (e.g., at lon = -8°), particularly for the ECMWF dataset. This increased variability can be attributed to the diurnal evolution of the boundary layer: the data include all BL tops from 06:00 to 18:00 local time. Since the boundary layer over land grows and decays throughout these hours, typical for continental and desert areas, averaging over this period naturally results in large standard deviations. A similar behavior is observed in the CALIPSO retrievals, which also show substantial variability above land. It is also worth noting that CALIPSO tends to detect lower BL tops than ECMWF over land. This difference likely arises from the way to define the BL top: ECMWF relies on thermodynamic criteria, while CALIPSO identifies a decrease in aerosol concentration. Consequently, aerosols detected by CALIPSO are mostly confined within the mixed layer, whereas ECMWF's BL height may include the residual layer or even the entrainment zone above it.*

*In the nighttime plot(Figure 6-right), the retrieved BL tops are as expected significantly lower over land for both datasets. Over the ocean, the agreement between ECMWF and CALIPSO remains good. Over land, however, a different pattern emerges: the ECMWF dataset shows little variability but reports lower BL heights than CALIPSO, particularly further inland (lon ¿ −10°). This again can be explained by the use of thermodynamic criteria to identify the BL top in ECMWF. In contrast, CALIPSO often detects aerosols residing in the residual layer or within the stable nocturnal boundary layer, resulting in systematically higher BL than ECMWF. An additional factor to consider is the quality of the CALIPSO nighttime profiles. The CALIOP instrument has different signal-to-noise characteristics during day and night: while solar background noise degrades daytime profiles, nighttime profiles suffer from lower photon count rates, which makes them noisier, especially over land. This effect is consistent with our findings in Appendix A2, where the correlation between ECMWF and CALIPSO is low.*

*Overall, the two datasets show generally good agreement over the ocean, where both daytime and nighttime results are consistent. This aligns with the findings from section 3.1 (Area 1). The agreement is also stronger during the daytime compared to the nighttime, reflecting the limitations of the satellite nighttime measurements. Over land, however, discrepancies emerge due to the strong diurnal cycle and the different methodologies used to define the BL top.*

2. Correlations between CALIPSO and ECMWF, PollyXT Lidar, and Halo Lidar over Cabo Verde (Section 3.3).

In Section 3.3 the authors compare CALIPSO BL height retrievals against ECMWF, PollyXT Lidar, Halo Lidar, and Radiosonde datasets over Cabo Verde. Even though the number of data points is rather small (e.g. 13 for CALIPSO collocations with PollyXT), this still would be an interesting opportunity to evaluate the strengths of different BL height measurement methods. Instead, the analysis part (lines from 261 to 274) is rather short and often seems inaccurate. The slopes of 0.66 and 0.63, in my view, do not indicate good agreement between datasets. There is no Halo lidar data that would suggest overestimation at lower values of BL height (the data fits are simply inconclusive). ECMWF does not retrieve BL height (it uses a parameterization based on vertical profiles of atmospheric parameters). It would be interesting to see how distance from ground observations (PollyXT, Halo, Radiosonde) and CALIPSO affects comparisons (the islands affect BL structure and CALIPSO measurements can be as far as 150 km away).

We thank the reviewer for the valuable comments. We have added the correlation coefficients to the plots to better represent the agreement between the datasets, rather than focusing on the slopes, and we have revised the discussion in Section 3.3 accordingly: *The correlation coefficient for PollyXT (red) and ECMWF (blue) lines, are r=0.69 and r=0.75 respectively, indicating that CALIPSO data present a rather satisfactory agreement with the model and the ground-based lidar. However, given their small positive intercepts (0.22 and 0.11), these datasets tend to estimate slightly lower BL compared to CALIPSO, even when their trends are generally aligned. The Halo lidar results, with the lowest correlation coefficient (r=0.37), show the weakest correlation with CALIPSO and the fit is inconclusive. The collocated cases may be limited, but suggest that CALIPSO generally captures the same variability in BL height as ECMWF and PollyXT, although with some systematic differences. The inconsistencies between Halo and CALIPSO BL results, reflect methodological differences, since Halo estimates the MLH from turbulence parameters while CALIPSO relies on gradient-based detection of layering. Similarly, ECMWF uses a thermodynamic approach (according to ECMWF ch.3), which may also contribute to discrepancies. The best agreement is found between the two aerosol lidars, highlighting that the choice of parameter used to define the BL height is critical for the assessment. CALIPSO and Polly use aerosols as tracers, identify-*

*ing the BL top from the sharp reduction in aerosol load at the transition to the free troposphere, whereas the Halo determines the BL height from turbulence, calculated through vertical velocity variance.* More-over, given the limited radiosonde data points collocated with the CALIPSO, we included a comparison between all the collocated radiosonde-PollyXT BL heights during the campaign (N=40, r=0.87). We also appreciate the suggestion to examine how the distance between CALIPSO overpasses and ground-based observations affects the comparisons. While this would indeed be an interesting analysis, given the extensive revisions already undertaken and the scope of the current manuscript, we consider addressing this aspect in future work.

3. Two case studies over Cabo Verde
The two cases presented in the manuscript are quite interesting and properly illustrate different inter-actions between the BL and free atmosphere. However, I find the atmospheric profiles and BL height measurements rather inconsistent in these two cases, and the authors do not provide a satisfying analysis and explanation of the datasets. The authors refer to the virtual potential temperature in Figs. 9a and 11a, but it clearly appears to be the regular temperature (it drops systematically with height within the BL). In the first case (dust above BL), the BL height is well characterized by radiosonde observations (Fig. 9a), but all the other datasets indicate lower, sometimes considerably, BL heights (Fig. 9b, c). Halo lidar measurements are almost half the radiosonde value. There is no attempt to reconcile these discrepancies. In the second case (desert dust within BL), the BL height determination is more com-plicated and the discrepancies between methods could be more justifiable. The radiosonde BL height should be included in both Figs. 9b and 11b.

We thank the reviewer for pointing this out. We have replaced the temperature profiles with the virtual potential temperature in Figures 9a and 11a, and we have included the radiosonde-derived BL in Figures 9b and 11b.
We do not consider the BL results inconsistent across the two case studies; rather, the differences highlight interesting features of the BL structure under different conditions. In the first case, the radiosonde indicates a relatively deep layer, while the lidars and the model show a shallower layer. This apparent discrepancy can be explained by the presence of a shallow layer (approximately 750–1000m) beneath a residual or elevated inversion around 1km, resulting from either large-scale forcing or the previous day's atmospheric structure. The lidar detects the top of the aerosol layer, whereas the radiosonde responds to the thermodynamic inversion top, which may be displaced horizontally during the ascent. Radiosondes drift with the wind during as they lift and can sample air parcels several kilometers away, potentially encountering a different boundary layer structure. In contrast, the lidar provides a vertically local measurement. Strong wind shear or the presence of an entrainment zone can also create layered aerosol structures that do not necessarily coincide with the thermodynamic layer top.
At the first case, the NNE wind was very strong (up to 14m/s), directing the radiosonde toward Monte Cara, which reaches an altitude of 490m (see map below). We believe that the discrepancy is primarily due to this horizontal displacement and does not represent the boundary layer structure directly above the Observatory. In the second case, however, the radiosonde ascends through the central part of the island that does not have orography, and results in a closer agreement with the lidar measurements.

We have revised and enriched the discussion of sections 3.3.1 and 3.3.2, and we hope that this analysis satisfactory explains the datasets and the discrepancies that arise.

[Figure]

Figure 1: Maps with the trajectory of the radiosonde balloon during its ascent for two case studies. Left: 12 September 2022 and Right: 23 September 2022. The color along the trajectory indicates the altitude of the radiosonde.

---

## Editor Decision (ED1)

I have reviewed the revised version of this manuscript, and especially the author response to reviewer comments. First, I would like to thank the reviewers for a thorough review and several very relevant comments. Second, I find that the authors have responded appropriately to these comments and that this have substantially improved the quality of the paper. Third I would like to apologize for lengthening and already long process; it was unusually difficult to find reviewers for this paper.

I have, however, one remaining issue that I would like to have resolved before I accept this paper for publication. This deals with an – as I believe – insufficient discussion on the key feature that this paper deals with; the atmospheric boundary layer and its depth.

Observing the boundary layer depth – or the height of the boundary-layer top – from space is a very timely issue; having a global climatology of the from space would open up a new chapter in boundary layer meteorology. This is also pursued in Nasa's Decadal Survey Incubation program and the NASA PBL Study Team (see DOI: 10.1175/BAMS-D-23-0228.1). In the light of this it would be important to discuss the fundamental problem: What is a boundary layer and how can its characteristics be estimated from space?

The text does an excellent job of describing the technical challenges with different metrics but it never clarifies these issues, which I believe makes the interpretation difficult. For example, I believe that the attempts by the authors to explain the differences between model and observations for the case studies by heterogeneity and sonde balloon drift are less than convincing, misguided and maybe even misleading. I think that the reasons instead lie in the fact that the authors are comparing apples and pears.

The atmospheric boundary layer by definition is the layer of the lowest of the atmosphere closest to and in direct contact with the Earth's surface, where mixing is maintained by turbulence. This cannot be directly simulated by models and hence not by ERA5; instead it is parameterized. Therefore, in ERA5 this layer is diagnosed from boundary-layer theory using a version of the critical Richardson number, Ric. None of this can be observed, neither from space, nor from surface based lidar and not from radiosondes. Instead different proxies are used; most commonly some kind of mixing concept often involving thermal structure, e.g. identifying inversions in temperature or moisture; sometimes also using aerosols.

In this context it is necessary to realize that just because the thermodynamic profiles suggests mixing has happened doesn't mean it is still ongoing. Both in the context of the residual layer and for decoupled cloud layers, a layer with seemingly well mixed potential temperature may be much deeper than the actual boundary layer as defined using a critical Ric. In such cases the inversion in potential temperature may not be the top of the boundary layer (cf. e.g. DOI:10.1002/2017JD027234) and the definition of it becomes a matter of choice. If the vertical gradient of the wind speed goes to zero at a lower height, Ri > Ric which will indicate a shallower boundary layer than the (main) inversion. Also, aerosols may remain unchanged in a residual layer, whereas in the actual boundary-layer it is affected by deposition, chemistry or clouds.

What I'm looking for here is not a solution to this problem, because there may not be one. I'm asking for an insightful paragraph or maybe just a few lines discussing this, acknowledging that differences between different methods and different instruments and methods may not indicate that one or the other is correct and the other wrong; it may just be that they measure different thing, none of which may be the actual boundary layer.

---

## Author Response (AR2)

First of all, we would really like to thank the Editor for the dedicated time and for the chance to further improve the scientific quality of the manuscript. In the following text, we have included the response to all comments, together with a description of the corresponding changes made in the revised manuscript. For clarity, the editor's comments are presented with black and responses with **blue**. All line numbers mentioned refer to the uploaded track-changes version of the manuscript.

I have reviewed the revised version of this manuscript, and especially the author response to reviewer comments. First, I would like to thank the reviewers for a thorough review and several very relevant comments. Second, I find that the authors have responded appropriately to these comments and that this have substantially improved the quality of the paper. Third I would like to apologize for lengthening and already long process; it was unusually difficult to find reviewers for this paper.

I have, however, one remaining issue that I would like to have resolved before I accept this paper for publication. This deals with an — as I believe — insufficient discussion on the key feature that this paper deals with; the atmospheric boundary layer and its depth.

Observing the boundary layer depth – or the height of the boundary-layer top – from space is a very timely issue; having a global climatology of the from space would open up a new chapter in boundary layer meteorology. This is also pursued in Nasa's Decadal Survey Incubation program and the NASA PBL Study Team (see DOI: 10.1175/BAMS-D-23-0228.1). In the light of this it would be important to discuss the fundamental problem: What is a boundary layer and how can its characteristics be estimated from space?

We have modified the introduction and included the following text (Lines 31-44), to support the discussion on the boundary layer and the fundamental problem of how its characteristics can be estimated from space:

"Over the open Atlantic, the Marine Atmospheric Boundary Layer (MABL) is typically shallow and influenced by the relatively constant sea surface temperature, while boundary layers in coastal and island regions experience terrestrial-marine interactions that increase their variability (Garratt, 1994; Wood, 2012). A limited number of studies over years have addressed the detection and analysis of MABL using lidar data, primarily due to practical and observational challenges over the ocean(e.g. Atlas et al. 1986; Flamant et al. 1997; Pena et al. 2015). Given these constraints, satellite observations can provide an important means of obtaining information in remote regions lacking in-situ and ground-based remote sensing data, while also enabling the development of global climatologies (Teixeira et al., 2025).

Although the BL is a near-surface phenomenon, several satellite measurements can indirectly infer its properties, particularly its depth and spatial or temporal variability. The Cloud-Aerosol

Lidar and Infrared Pathfinder Satellite Observations (CALIPSO) mission, has been widely used to derive global BL height climatologies over ocean and land and is therefore essential for studying lower troposphere characteristics (e.g. Liu et al. 2024). Nevertheless, when interpreting satellite-derived BL characteristics, it is crucial to decode the measurements appropriately, as the definition and identification of the BL can vary depending on the chosen approach and physical parameter. The Cloud-Aerosol Lidar with Orthogonal Polarization (CALIOP) of the CALIPSO satellite, can measure, among others, backscattered light from aerosols and clouds. Hence, in this case, the top of the lowest aerosol layer often coincides with the BL top, since aerosols are typically well mixed within the BL and drop sharply above it (Li et al., 2017)."

The text does an excellent job of describing the technical challenges with different metrics but it never clarifies these issues, which I believe makes the interpretation difficult. For example, I believe that the attempts by the authors to explain the differences between model and observations for the case studies by heterogeneity and sonde balloon drift are less than convincing, misguided and maybe even misleading. I think that the reasons instead lie in the fact that the authors are comparing apples and pears.

We agree that the apparent discrepancies between model and observational estimates largely arise because the different approaches represent distinct physical aspects of the boundary layer. While we had noted this point in the manuscript (see below: lines 432–434 and 455–456), we have now expanded the discussion of the first case and the conclusions. We hope that this addition explicitly acknowledges that the different methods may not measure the same quantity, which explains the observed differences without implying that one approach is necessarily more accurate than another.

lines (432-434) "The differences observed across the instruments largely reflect the distinct definitions and retrieval methods used to estimate the BL top, emphasizing that no single dataset provides a complete picture on its own."

lines (455-456): "Hence, the height detected for the BL top needs careful treatment and the interpretation is highly dependent on the definition and methods used."

We have changed the discussion at the end of the section 3.3.1 (Lines 379-383):

"Upon assessing all the BL results together, we find that the two lidars are in good agreement, consistently capturing the well-mixed aerosol layer. In contrast, the radiosonde indicates the strongest inversion at around 1 km, which is relatively high for a BL in this region and differs significantly from the lidar results. As discussed by Brooks et al. (2017), an apparently well-mixed potential temperature profile may extend into a residual layer where turbulent mixing is no longer active, leading to an overestimation of the actual BL height. Moreover, while lidar detects the top of the aerosol mixing layer, radiosondes diagnose stability changes that may reflect remnants of

earlier mixing. Therefore, these differences between lidar- and radiosonde- derived boundary layer heights can be expected, particularly under conditions of weak turbulence or decoupled layers."

We have also enriched the explanation of how BL top is derived from ECMWF dataset (Lines 141-146).

The atmospheric boundary layer by definition is the layer of the lowest of the atmosphere closest to and in direct contact with the Earth's surface, where mixing is maintained by turbulence. This cannot be directly simulated by models and hence not by ERA5; instead it is parameterized. Therefore, in ERA5 this layer is diagnosed from boundary-layer theory using a version of the critical Richardson number, Ric. None of this can be observed, neither from space, nor from surface based lidar and not from radiosondes. Instead different proxies are used; most commonly some kind of mixing concept often involving thermal structure, e.g. identifying inversions in temperature or moisture; sometimes also using aerosols.

In this context it is necessary to realize that just because the thermodynamic profiles suggests mixing has happened doesn't mean it is still ongoing. Both in the context of the residual layer and for decoupled cloud layers, a layer with seemingly well mixed potential temperature may be much deeper than the actual boundary layer as defined using a critical Ric. In such cases the inversion in potential temperature may not be the top of the boundary layer (cf. e.g. DOI:10.1002/2017JD027234) and the definition of it becomes a matter of choice. If the vertical gradient of the wind speed goes to zero at a lower height, Ri > Ric which will indicate a shallower boundary layer than the (main) inversion. Also, aerosols may remain unchanged in a residual layer, whereas in the actual boundary-layer it is affected by deposition, chemistry or clouds.

What I'm looking for here is not a solution to this problem, because there may not be one. I'm asking for an insightful paragraph or maybe just a few lines discussing this, acknowledging that differences between different methods and different instruments and methods may not indicate that one or the other is correct and the other wrong; it may just be that they measure different thing, none of which may be the actual boundary layer.

We have modified the conclusions as following (Lines 444-453):

"It is important to note that differences between the BL heights derived from different instruments/model and methods do not necessarily imply that one is correct and the other is wrong. Rather, they often reflect the fact that each technique responds to a different physical aspect of the boundary layer. For instance, in ERA5, the BL top is not explicitly resolved but diagnosed from boundary-layer theory using a critical Ric, representing the depth of active turbulent mixing. Radiosonde-derived heights are typically based on thermodynamic structure, identifying the strongest temperature or humidity inversion, which may correspond to a residual

layer rather than the actively mixed layer. Lidar measurements, in contrast, detect gradients in aerosol backscatter, which trace the extent of aerosol mixing but may remain unchanged even after turbulence ceases. An apparently well-mixed potential temperature profile may thus extend well above the dynamically defined boundary layer (Brooks et al., 2017). Therefore, the discrepancies observed between model, lidars, and radiosonde estimates likely arise because these approaches describe related but not identical layers within the lower atmosphere."

---

## Author Response (AR3)

**Authors' Response**

We would like to thank all those involved for their time and effort in reviewing, commenting on, and contributing to this manuscript. The Editor and reviewers have provided valuable feedback that has significantly improved the scientific quality of the paper. Below, we would like to state the following points (points 1,2) and report some relevant changes made in the manuscript (points 3,4).

- 1. The full names for all authors are used.
- 2. I ensure that the reproduction rights for all figures have already been secured and that maps and aerials include the required copyright statements or credits as requested by the providers.
- 3. Regarding Area 1, the figure 5 left is improved, with equally spaced conceptual day and night trajectories of CALIPSO.
- 4. The datasets that are used for the results of this work can be found in the data availability section: "The ASKOS Campaign dataset is available from the ESA Atmospheric Validation Data Centre (EVDC) at https://evdc.esa.int with DOI: 10.60621/jatac.campaign.2021.2022.caboverde \citep{marinou-2023-askos}. Visualized datasets of the ASKOS Campaign and additional information are also available at https://askos.space.noa.gr/data; The ERA5 reanalysis dataset used in this study are available from the Copernicus Climate Change Service (C3S) Climate Data Store: \citeauthor{era5}: ERA5 hourly data on single levels from 1940 to present, (C3S CDS), https://doi.org/10.24381/cds.adbb2d47. The livas dataset is available upon request."